# Optical Properties of Polyisocyanurate–Polyurethane Aerogels: Study of the Scattering Mechanisms

**DOI:** 10.3390/nano12091522

**Published:** 2022-04-30

**Authors:** Beatriz Merillas, Judith Martín-de León, Fernando Villafañe, Miguel Ángel Rodríguez-Pérez

**Affiliations:** 1Cellular Materials Laboratory (CellMat), Condensed Matter Physics Department, Faculty of Science, University of Valladolid, Campus Miguel Delibes, 47011 Valladolid, Spain; jmadeleon@fmc.uva.es (J.M.-d.L.); marrod@fmc.uva.es (M.Á.R.-P.); 2GIR MIOMeT-IU Cinquima-Química Inorgánica, Faculty of Science, University of Valladolid, Campus Miguel Delibes, 47011 Valladolid, Spain; fernando.villafane@uva.es; 3BioEcoUVA Research Institute on Bioeconomy, University of Valladolid, 47011 Valladolid, Spain

**Keywords:** polyurethane–polyisocyanurate aerogels, optical properties, transparency, Rayleigh scattering, Mie scattering, scattering mechanisms

## Abstract

Highly transparent polyisocyanurate–polyurethane (PUR–PIR) aerogels were synthesized, and their optical properties were studied in detail. After determining the density and structural parameters of the manufactured materials, we analyzed their optical transmittance. It was demonstrated that the catalyst content used to produce the aerogels can be employed to tune the internal structure and optical properties. The results show that the employment of lower catalyst amounts leads to smaller particles forming the aerogel and concomitantly to higher transmittances, which reach values of 85% (650 nm) due to aerogel particles acting as scattering centers. Thus, it was found that the lower this size, the higher the transmittance. The effect of the sample thickness on the transmittance was studied through the Beer–Lambert law. Finally, the scattering mechanisms involved in the light attenuation were systematically evaluated by measuring a wide range of light wavelengths and determining the transition between Rayleigh and Mie scattering when the particles were larger. Therefore, the optical properties of polyurethane aerogels were studied for the first time, opening a wide range of applications in building and energy sectors such as glazing windows.

## 1. Introduction

Aerogels are open-porous materials obtained when the liquid solvent filling the gel pores is replaced by air while the porous structure is kept almost intact. These materials present a growing interest owing to their exceptional properties, i.e., high porosities (80–99%), ultralow densities (3–500 kg/m^3^), low thermal conductivities (10–100 mW/mK), and high specific surface areas, among others [1]. 

The combination of these exceptional properties has awakened a great level of interest in the production of aerogels from a great variety of materials. In particular, some of these aerogels show a significant transmittance to visible light, with some applications of aerogels based on the combination of two of the most stimulating properties, i.e., their low thermal conductivity and their high light transmittance [2,3,4,5,6]. For this reason, transparency is an increasingly demanded property for aerogels.

Transparency in aerogels is usually linked to silica matrices, and in fact, silica aerogels commonly present high transmittances [7,8,9]. The fundamental explanation of this high transparency is, on the one hand, the negligible light absorption of the silica matrix, unlike carbon- or metal-based aerogels [10,11]. On the other hand, these structures slightly scatter visible light. The size of the particles forming the silica aerogel is smaller than a tenth of the visible wavelength, which leads to a small amount of scattered light and, therefore, to a high transmittance. This behavior can be described through Rayleigh’s scattering [12], which states an inverse relationship between the transmitted intensity and the fourth power of the wavelength, which explains the characteristic bluish color of some aerogels [13]. Nevertheless, when the size of the scatterers is similar to the wavelength of light, Mie scattering takes place, and the transmitted intensity varies with the first power of the light wavelength [14]. The Rayleigh-like behavior has been widely studied in the literature. For example, Mandal et al. [15] analyzed the haze of silica wet gels and aerogels presenting low densities (0.017 g cm^−3^), which were attributed to Rayleigh scattering from secondary particles rather than from pores. They explained how scattered light also depended on the initial sol concentration and, therefore, on the final density. Zhao et al. [16] developed a theoretical model for the optical performance of aerogels. This model, based on the radiative transfer equation, demonstrates that the radiative properties depend only on the aerogel microstructure. Experimental data were used to validate this proposal. With this purpose, absorption and scattering were separately quantified, obtaining for the latter a dependence on a wavelength of λ^−4^ (typical of Rayleigh scattering). 

Additionally, different features affecting the final light transmittance have been widely studied in the literature. Using ultra-small angle X-ray scattering measurements, Emmerling et al. [17] explained how the scattered light depends on nanostructural parameters of the gel network, such as the average particle size, interparticle voids, pore size, and an ordering factor related to the monomer concentration. Twej et al. [18] studied the relationship between the pH during the synthesis of silica aerogels, their morphology, and their final optical properties. In this way, a strong dependence was found of the light transmittance on the wavelength, which was governed by the microstructure features. 

Although silica aerogels have high transparency, it is highly interesting to produce organic aerogels with high transparency due to their potential improved mechanical performance [e.g., higher toughness) than inorganic aerogels [19]. The possibility of increasing the hydrophobicity and the ease of achieving a more controlled and predictable nanostructure [20] are other features favoring organic aerogels. There are several examples of polymeric aerogels showing high light transmittances such as melamine-formaldehyde aerogels [21], chitosan-based aerogels [22], nanocellulose aerogels [23], polyimide aerogels [24], or even composites based on cellulose-nanofiber–polysiloxane aerogels. 

In particular, the use of transparent aerogels based on polyurethane/polyisocyanurate could lead to the combination of a transparent material showing a very low thermal conductivity and a cost-effective solution based on a well-known polymer matrix in the building sector. Recently, the first transparent PUR–PIR aerogels were synthesized by obtaining high values of light transmittance [25]. Herein, we present a detailed and systematic study of the optical properties, as well as the effect of different factors (aerogel thickness, porous structure, and incident light wavelength) on the optical transmittance of these PUR–PIR aerogels. In addition, the scattering mechanisms in these aerogels were analyzed. 

Therefore, the microstructure–properties relationship is assessed through the analysis of the structure and morphology, as well as the study of nanoscale physics phenomena, for these ultralightweight aerogels.

## 2. Materials and Methods

### 2.1. Materials

IsoPMDI 92140 (p-MDI) (ρ = 1.23 g/cm^3^) was supplied by BASF Polyurethane. Pentaerythritol (PTOL) (ρ = 1.396 g/cm^3^) was purchased from Alfa Aesar (Haverhill, MA, USA). A catalyst of potassium octoate (KOSMOS 75 MEG) was obtained from Evonik (Essen, Germany).

Acetonitrile (purity > 99.9%), DMSO (purity > 99.5%), and tetrahydrofurane (purity > 99.5%) stabilized with 250 ppm butylated hydroxytoluene were provided by Scharlab, S. L. (Barcelona, Spain).

### 2.2. Aerogels Production

Polyisocyanurate–polyurethane (PUR–PIR) aerogels were prepared by a sol-gel technology using the following procedure, previously described by the authors [25]: a solution of isocyanate (p-MDI) (44 g/L in CH3CN:THF at 65:35%vol.) was poured into a plastic cup, and then the corresponding amount of the pentaerythritol solution (PTOL) (100 g/L in DMSO) was added to obtain a molar relationship of PTOL/p-MDI of 0.43. Finally, a catalyst (2–18 wt.% of the total of PTOL and p-MDI mass) was quickly added to the mixture, and the solution was stirred at 500 rpm for 20 s at room temperature (controlled temperature at 22 °C). The obtained sol was poured into a plastic syringe (2 mm in diameter) until the gelation time was reached. Then, a 24 h aging step and two washing steps (of 24 h each) were performed by covering the gels with pure acetonitrile to remove residual reagents. Syringes were tightly sealed with a plastic film in order to avoid solvent evaporation and premature drying of gels. Finally, gels were demolded by carefully pressing the syringe plunger. Then, gels were covered with acetone to prevent evaporation of the solvent and were dried by SCD-CO_2_ (supercritical drying) at 40 °C and 100 bar. The drying time depends on the number of samples placed into the autoclave and the volume thereof. Cylindrical samples were manufactured by keeping the same amount of p-MDI and PTOL but employing different amounts of catalyst (ranging from 2 to 18 wt.%) with dimensions of ca. 16 mm in diameter and 10 mm in height. These samples were cut with a metal blade to obtain different thicknesses for the measurements of light transmittance.

### 2.3. Characterization Techniques

#### 2.3.1. Density and Porosity

The geometrical density of the obtained cylindric aerogels was calculated by measuring their mass with an AT261 MettlerToledo (Columbus, OH, US) balance and their volume with a caliper, as described in ASTM D1622/ D1622M-14 [26]. The outer skin (ca. 2 mm) of the produced specimens was removed for those measurements. 

Porosity (Π) was calculated as described by the following equation (Equation (1))
(1)$$\Pi =\left(1-\rho_{r}\right)*100$$
where ρ_r_ is the relative density, defined as
(2)$$\rho_{r}=\frac{\rho }{\rho_{s}}$$
where ρ is the geometric density of the aerogel and ρ_s_ is the solid density of the polymeric matrix, i.e., 1.17 g/cm^3^, determined by helium pycnometry [25]. 

#### 2.3.2. Scanning Electron Microscopy

Aerogel samples were cut and metalized through an iridium sputter coater (EMITECH K575X Sputter Coater, Montigny-le-Bretonneux, France) to avoid altering their porous structure [27]. Scanning electron micrographs were obtained using an ESEM Scanning Electron Microscope (QUANTA 200 FEG, Hillsboro, OR, USA).

#### 2.3.3. Specific Surface Area

The nitrogen sorption technique was used for measuring the specific surface area of the aerogel samples with the Brunauer–Emmett–Teller (BET) method [28]. After a sample degasification step under a high vacuum (24 h at 50 °C), measurements were made at −196 °C in the range P/P_0_ = 0.05–0.30. These measurements were carried out with a Micromeritics (Norcross, GA, USA) ASAP 2020 instrument at the University of Málaga (Málaga, Spain).

#### 2.3.4. Particle and Pore Size

The particle size was determined using software based on Image J (2.0-1.52q, 2019)/FIJI [29] and SEM micrographs. The visualized particles were drawn and their diameter was estimated. The particle size is the mean value of at least 50 particles. The standard deviation and particle size distribution were also measured for each formulation. 

The pore size was estimated along the N_2_-desorption isotherm using the Barrett–Joyner–Halenda (BJH) method [28]. Due to capillary condensation occurring for small pores, larger pores could not be measured. Thus, the total pore volume (V_p_) was calculated using the following equation
(3)$$V_{p}=\frac{1}{\rho }-\frac{1}{\rho_{s}}$$
where ρ is the geometric density of the aerogel and ρ_s_ is the solid density of the polymeric matrix. In addition, considering the specific surface area and the total pore volume, the average pore size can be determined using Equation (4)
(4)$$\Phi_{p\;}=\frac{4V_{p}}{S_{\mathrm{BET}}\;}$$

#### 2.3.5. Transmittance Measurements

Light transmittance was calculated as the ratio between the intensity passing through the aerogel sample (I) and the intensity without sample (I_0_). These measurements were carried out by employing three lasers: a red laser (650 nm) and a green laser (532 nm), provided by Laserlince S.L. (Valladolid, Spain), and a blue laser (450 nm) model MLL-III-450L supplied by Lasing S.A (Madrid, Spain). The complete transmittance measurements system comprises the laser acting as light source and a photodiode joined to an integrating sphere with 12.5 mm window (PRW0505, Gigahertz-Optik) connected to a photometer (X94, Gigahertz-Optik, Türkenfeld, Germany) as a light detector (detector–laser distance fixed at 133 mm). Samples were placed directly in the window of the integrating sphere to collect all the light scattered by the samples. A thin slide of ca. 2 mm was removed from both cylinder base surfaces of the samples to maintain a homogeneous structure and avoid surface effects. Different thicknesses have been measured for each sample by cutting plane-parallel slices with a metal blade (from 0.4 mm to 12 mm).

#### 2.3.6. UV–Vis

An ultraviolet spectrometer (UV-2102 PC, Shimadzu, Kyoto, Japan) was used to evaluate the wavelength dependence of the transmitted light in the range from 400 to 900 nm. The experiments were performed with an aperture of the slit of 1 nm. A background spectrum without any sample was made to take this intensity as 100% of transmittance. According to the UV–Vis assembly, samples were placed 2 cm from the detector, meaning that some of the transmitted light did not reach the detector. For this reason, UV–Vis transmittance values were taken as relative values by comparison between samples and as a strategy to determine the dependence of the wavelength on the transmitted intensity.

## 3. Results and Discussion

### 3.1. Aerogels Structure

SEM images were taken (Figure 1) in order to analyze the porous structures of the samples under study (micrographs with higher magnification are included in the Appendix A). On the one hand, it can be seen that the obtained aerogels present a homogeneous nanostructure, which is relevant for obtaining high transparency values. On the other hand, the aerogel nanostructures vary with different amounts of catalyst, showing significantly different particle and pore sizes. The images indicate that increasing the catalyst content leads to larger particle and pore sizes. This effect was quantified by analyzing the particle size distribution and the mean particle size.

The particle size distribution of each sample affects the efficiency of the scattering mechanism. Figure 2 displays the particle size distribution for the content of each catalyst. Narrow distributions are obtained for those aerogels with a low catalyst content in their formulation (2, 3, 4, and 6 wt.%), which are formed by particles with a homogeneous size. In fact, almost all particles have sizes below 40 nm in these aerogels. Nevertheless, for samples containing higher catalyst amounts (8, 10, 15, and 18 wt.%), the size of the polymeric particles is more heterogeneous, covering a wider range of the nanometric scale, with a significant number of particles larger than 40 nm.

These distributions affect the scattering mechanisms, as discussed in the next section. Table 1 shows the structural parameters describing the aerogel structures, including the average particle size obtained from the particle size distributions.

The particle sizes range from 23 to 78 nm and show a clear increase with the catalyst amount, as previously discussed. Values above 40 nm, a tenth of the wavelength of the visible light, were only reached for the two samples with higher amounts of catalyst: 15 and 18 wt.%. Regarding the mean pore sizes, the same tendency is observed with the catalyst content with pore sizes that increase from 85 to 141 nm for the samples with catalyst contents ranging from 2 to 10 wt.%. A sharp increase in the average pore size was obtained for the samples with the highest catalyst content, reaching a maximum value of 722 nm. The explanation of these differences is related to the reaction kinetics that control the polycondensation reaction in which the polymeric skeleton is formed [25]. Therefore, when the solid network is slowly formed (a low catalyst content), a higher number of small particles is present. On the other hand, when the catalyst amount is increased, promoting a faster skeleton formation, particles form aggregates that give rise to larger secondary particles.

The structures described above lead to different density values, ranging from 0.079 g/cm^3^, for the material produced using the highest catalyst content, to 0.128 g/cm^3^, for the material produced using a catalyst content of 3 wt.%. Porosity values are in a narrow range, from 89 to 93%, indicating that a large gas volume fraction is contained in these aerogel samples. The values for the structural parameters obtained by nitrogen sorption (pore volume and specific surface areas) and the nitrogen adsorption–desorption curves can be found in Appendix A, respectively.

### 3.2. Optical Transmittance

#### Effect of Catalyst and Thickness

Transmittance is a measure of the amount of light that is able to pass through a material in comparison with the incident light. However, far from the simple appearance of this definition, many factors and parameters affect the light transmitted by aerogels. In the following sections, these key parameters are discussed.

For all the analyzed samples, the dependence of the transmittance on the thickness of the samples is displayed in Figure 3. This relationship was analyzed at three different wavelengths: 450, 532, and 650 nm.

Transmittance shows the same behavior for the three laser wavelengths measured: the lowest catalyst contents lead to aerogels with higher transmittance values. The highest values, above 85% of transmittance, were reached for 650 nm (red color); around 78% for 532 nm; 65% for 450 nm: the absolute value clearly decreased when the light wavelength was reduced for each sample and each thickness. The samples produced with larger catalyst contents show very low values of transmittance, below 7%, even for very low thicknesses.

### 3.3. The Beer–Lambert Law

Regarding the sample thickness in Figure 3, the transmittance follows an exponential decay for samples between 2 and 10 wt.%, which follows the Beer–Lambert law. High transmittances are still obtained for aerogels under 4 mm in thickness, but the values are clearly smaller for thicker aerogels. 

According to the Beer–Lambert law [30] (Equation (5)), light transmission is related to the attenuation coefficient (µ) and the sample thickness (L):(5)$$T=\frac{I}{I_{0}}=e^{-\text{µ}\cdot L}$$

Therefore, if −ln T is represented as a function of the aerogel thickness, a straight line should be obtained whose slope corresponds to the attenuation coefficient µ. The obtained transmittance data are presented using this strategy in Figure 4 for the three light wavelengths used, for all the catalyst amounts, and for the different thicknesses.

Paying attention to the previous graphs, two groups of samples can be differentiated: the first one comprises the samples between 2 and 10 wt.% of catalyst contents and shows a good fit for the linear regression and similar behavior, whereas the second group, formed by samples with 15 and 18 wt.% of catalyst contents, presents a steeper slope and differs from the linear behavior. The adjusted R^2^ values for the fitting can be found in the Appendix A. Owing to the large size of the scatterers that these two samples present (15 and 18 wt.%), the trend of the fitting does not show a transmittance (T) value of 1 when the thickness (L) is 0. This is mainly due to very low values of the transmitted intensity for these samples, which makes the measurements less accurate. Despite the slight lack of accuracy of some measurements, it can be concluded that all the samples follow the Beer–Lambert law. This is an essential requirement to obtain the transmitted intensity values for any desired thickness from the experimental data, as described in Equation (6), which is derived from the Beer–Lambert law:(6)$$T=T_{0}^{\frac{L}{L_{0}}}$$
where T_0_ corresponds to a reference transmittance value, L_0_ is the real sample thickness, and L is the selected thickness to calculate the transmittance.

As described in Equation (5), light transmittance is related to the attenuation coefficient (µ). This parameter accounts for the scattered and absorbed light, i.e., the light that does not reach the detector. It can be calculated from the slope of the −ln vs. L plot (Figure 4). The obtained attenuation coefficients for the selected light wavelengths have been normalized with the sample density, as shown in Figure 5a, in order to avoid its influence, as demonstrated by Thiagarajan et al. [31].

This parameter takes into account all the factors affecting the reduction in the transmittance, such as the particle size and the light wavelength; these dependences are analyzed in Section 3.4.2.

In order to compare the optical properties of the samples, transmittance values for different catalyst amounts for a constant thickness of 1 mm were calculated by Equation (6). As expected, µ reaches the highest values for the samples with higher catalyst contents and, therefore, less transparent samples. Moreover, these results are in agreement with the transmittance for the different wavelengths (Figure 5b) since the attenuation coefficient is reduced as the wavelength increases, as discussed below.

### 3.4. Effect of the Aerogel Structure on Transmittance

The obtained transmittance results and the observed tendencies can be explained by considering the light interaction with the aerogel internal structure. When light passes through an aerogel, two processes can contribute to attenuating the radiation: scattering and absorption. Thus, part of the incident radiation may be transformed into thermal energy, corresponding to an absorption process that, although it depends on the scattering mechanisms [12], will be neglected in this work since all the aerogels present the same polymer matrix and a similar relative density. Additionally, the scattering mechanism deviates the incident light with a certain angle depending on the characteristic features of the scatterer centers.

Aerogels are open-porous materials formed by interconnected voids and a network of polymeric nanoparticles, i.e., a continuous air phase in which interconnected particles are dispersed. Therefore, in terms of its light transmission, the aerogel can be considered a continuum air medium through which the light can pass through, being scattered by the presence of nanometric particles. Moreover, the aerogel presents a refractive index slightly different from air, so the transmitted light would be slightly reduced by the refraction caused by the air–aerogel interface. The following sections analyze this effect and the scattering produced by the particles.

#### 3.4.1. Refraction Index

Due to the different refractive index between the polymeric matrix and the air, there is an inhomogeneous interaction between light and the aerogel [6]. Since more than 89% of the aerogel (Table 1) is air, its refractive index is expected to be very close to 1, meaning that light will only be slightly refracted by the PUR–PIR aerogels. Although the refractive index depends on the bulk density of the samples, these density values are within a narrow range (0.079–0.128 g/cm^3^), so we can assume that the refraction index will be similar for all the aerogels under study. The refraction index value can be calculated for each sample by the Clausius–Mosotti formula (Equation (7)) [32]:(7)$$n-1=\frac{3\;\rho }{2{\;\rho }_{s}}\;\left(\frac{n_{s}^{2}-1}{n_{s}^{2}+2}\right)$$
where the solid density (ρ_s_) for polyurethane is 1.160 g/cm^3^ and the refractive index for solid polyurethane (n_s_) is 1.67.

The refraction indexes thus obtained for our samples range between 1.038 and 1.062 (Appendix A), and thus, since they are close to 1, there is no significant refraction in the interface.

#### 3.4.2. Scattering Produced by Particles

Aerogels’ transparency mainly depends on the size of the structural inhomogeneities (i.e., particles forming the solid skeleton), the particle size distribution, and the packaging of these inhomogeneities, i.e., the aerogel density [6]. Particles act as nanometric scattering centers for the incident radiation by radiating electromagnetic energy in all directions. Therefore, the size of the structural elements forming the solid 3D network plays an essential role in the amount of scattered light, as well as in the direction the light is scattered. When the incident wavelength is similar to the scatter size, Mie scattering takes place [14,33]. In this case, most of the light is scattered in the forward direction. When the scatter size is reduced up to a tenth of the incident light, scattering turns into Rayleigh scattering [34]. In this case, the amount of scattered light is reduced, leading to a higher transmittance, whereas the wavelength dependence becomes stronger.

The aerogels’ density depends on the packaging of the spherical particles and the shrinkage experimented by the samples during the supercritical drying step. Thus, this factor must be considered in order to normalize the final transmittance, thus compensating for the potential deviations associated with different densities. The normalized transmittance (T/ρ) allows the effect of particle size to be analyzed more independently and enables a reliable comparison between different samples. The relationship between the normalized transmittance (for aerogels with 1 mm thickness) and the sizes of the scattering centers are displayed in Figure 6. The normalized transmittance shows a strong dependence on particle size, even for a narrow range of particle sizes (23–78 nm). The normalized transmittance is sharply reduced when the particle size increases from 23 to 30 nm. For particle sizes above 30 nm, the increase in the particle size leads to small variations in transparency. This fact is in agreement with the well-known transparency reduction occurring when the heterogeneities have a larger size than a tenth of the visible-light wavelength [35] (400–700 nm).

Thus, a slight increase in the final particle sizes leads to a significant loss in visible-light transparency. Hence, careful control of the reaction kinetics during the sol-gel process and an optimized supercritical drying method that avoids huge shrinkages and structural damage are crucial in the synthesis of transparent aerogels. PUR–PIR aerogels containing a low catalyst percentage, with particle and pore size values of a few nanometers, therefore display the highest transmittances.

### 3.5. Effect of the Light Wavelength

As previously discussed, the incident wavelength strongly affects the transmittance. Three different regions can be distinguished in the graph of Figure 5b regarding the effect of wavelength: the transmittance dependence on wavelength is less noticeable for samples with a high catalyst content (15 and 18 wt.%) since their particle sizes surpass 40 nm, i.e., above a tenth of the wavelength of visible light. The decrease in transmittance observed for the rest of the samples can be explained by the light scattering that those particles produce. In particular, samples with 2, 3, and 4 wt.% present a strong dependence on the light wavelength since the diameter of their particles is below 30 nm. The last group is formed by samples with 6, 8, and 10 wt.%, which are composed of particles with diameters of ca. 30 nm; thus, the dependence on the wavelength is still strong.

This dependence was studied in detail through the results obtained using a UV–Vis spectrometer. This spectroscopic technique has been used to cover a larger range of light wavelengths (400–900 nm) and analyze the dependence of transmittance on the incident wavelength. Figure 7 shows the transmittance curves for all the samples under study (for 1 mm thickness), except for those with the highest catalyst content (15 and 18 wt.%), since the latter display very small transmittance and poor signal-to-noise ratio.

Figure 7 shows a clear trend between transmittance and the amount of catalyst; this trend is higher for samples with less catalyst, which is in agreement with the results of the previous sections. Moreover, the strong effect of the wavelength on the final transmittance (Figure 7) confirms the difficulty for light to travel through the aerogel samples at small wavelengths.

The effect of light wavelength on the optical properties has been previously analyzed for silica aerogels, and their transmittance behavior has been attributed to Rayleigh scattering [36,37,38]. Rayleigh scattering refers to the elastic scattering of light when interacting with scatterer centers with a diameter smaller than about a tenth of the incident light wavelength [34]. The scattered intensity from Rayleigh scattering, I, that particles produce can be described as follows [39]:(8)$$I_{\mathrm{scattered}}=\frac{8{\;\pi }^{4}d^{6}}{r^{2\;}\lambda^{4}}\left[\frac{n^{2}-1}{n^{2}+2}\right]\left(1+\cos^{2}\theta \right)$$
where d is the particle diameter, n is the refractive index of the scatterer center, r is the radial distance from the scatterer, and θ is the scattering angle. The dependence of the scattered intensity on the scatterer size is to the sixth power, which explains the notable differences in the final transparency when particles increase in size (Section 3.1). According to Equation (8) of Rayleigh’s law, the scattering intensity varies with the fourth power of the inverse wavelength of the incident light. Therefore, transmission can be expressed as follows [35]:(9)$$T={A\;e}^{-\frac{\mathrm{BL}}{\lambda^{4}}}$$
where A and B are constants and L is the sample thickness. 

Nevertheless, the scattering mechanism produced when the scatterer’s diameter is larger than about one-tenth of the wavelength is known as Mie scattering [14]. Maxwell’s equations must be solved under the scattering conditions [40,41]. For particles with a radius greater than about ten times the wavelength of the light, the scattered intensity is given by:(10)$$I_{s}=I_{i}\int_{\alpha_{0}}^{\alpha_{1}}{\left|S\left(\theta ,\alpha \right)\right|}^{2}f\left(\alpha \right)d\alpha$$
where I_i_ is the total transmission, S(θ, α) is the scattering function, and f(α) is the distribution for the particle size parameter (α). This parameter can be described as:(11)$$\alpha =\frac{2{\;\pi n}_{m}}{\lambda }r$$
where n_m_ is the refractive index, λ the wavelength, and r the particle radius.

The transmittance–wavelength dependence varies, in this case, with the first power of the inverse wavelength of the incident light. In this way, with −ln(T) representing a function of either L/λ^4^ or L/λ in the range of 400–900 nm, and analyzing whether the data fit better, considering one or the other dependence allows us to understand the scattering mechanism in each case (Appendix A).

A summary of the fit of the average of the adjusted R^2^ for the fitting is shown in Table 2 (all the data in Appendix A). The obtained values confirm that there are different scattering mechanisms in the PUR–PIR aerogels described herein. Samples are divided into the three groups discussed above based on their different behaviors (Figure 6).

The three samples with the lowest amounts of catalyst (2, 3, and 4 wt.%), where the particles’ diameter is clearly less than a tenth of the visible wavelength, fit significantly better with the relationship L/λ^4^, indicating that the Rayleigh scattering mechanism occurs. This leads to a smaller amount of scattered light, and, therefore, transmittance in these samples is notably high. Samples with 6 and 8 wt.% still show a Rayleigh regime. However, the particle size distribution of the aerogels with 10 wt.% of catalyst content is shifted to higher values, meaning that Mie scattering is the principal scattering mechanism (see the structural parameters shown in Table 1 and particle size distribution shown in Figure 2). This means that the number of scattered light increases and, therefore, that transmittance decreases, as demonstrated experimentally.

## 4. Conclusions

Transparent polyisocyanurate–polyurethane (PUR–PIR) aerogels were synthesized using different amounts of catalyst. These samples are characterized in terms of density, porosity, and porous structure. Scanning electron microscopy was used to observe their structures, and their pore and particle sizes were estimated. The particle size varies with the amount of catalyst, with values ranging from 23 to 78 nm. The particle size distributions indicate that larger particles lead to a more heterogeneous distribution.

The optical properties have been analyzed in detail. The dependence of the transmittance was studied as a function of different parameters. Regarding the catalyst content, the reaction kinetics during the formation of aerogels had a significant effect on their optical transmittance, with the samples with the highest transparency being the samples with the lowest catalyst content. On the other hand, aerogels were cut at different thicknesses (from 1 to 14 mm) to evaluate the effect on the final transmittance. The fitting of these data demonstrated the fulfillment of the Beer–Lambert law. Then, based on the dependence of transmittance on thickness, the attenuation coefficient was calculated for all the samples, and a noticeable trend was obtained, with the catalyst amount showing the dependence on the particle size and light wavelength.

Regarding particle size, a strong correlation was demonstrated between the normalized light transmittance and the size of the particles forming the different samples. Aerogels with an average particle size below 30 nm and narrow cell size distributions showed transmissions that were significantly higher than those with particle sizes larger than 30 nm and/or wider particle size distributions. 

Finally, the corresponding scattering mechanism was evaluated by analyzing the dependence of the transmittance on a wide range of light wavelengths (400 to 750 nm). The main scattering mechanism was obtained by fitting the transmittance to L/λ^4^ or L/λ by calculating the R^2^ values. It was demonstrated that Rayleigh scattering was the main mechanism present when particle sizes were between 23 and 32 nm, whereas Mie scattering was shown to begin taking place when the size of the aerogel particles increased above these values. This fact explains the loss of transparency when the particles forming the PUR–PIR aerogels exceed those values.

## Figures and Tables

**Figure 1 nanomaterials-12-01522-f001:**
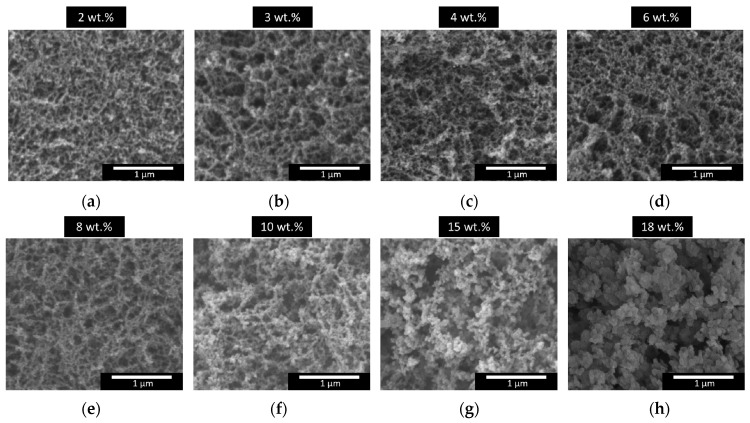
SEM Micrographs of the porous structures of polyisocyanurate–polyurethane (PUR–PIR) aerogels for different catalyst contents ranging from 2 to 18 wt.%, (**a**) 2wt.%, (**b**) 3 wt.%, (**c**) 4 wt.%, (**d**) 6 wt.%, (**e**) 8wt.%, (**f**) 10 wt.%, (**g**) 15 wt.% and (**h**) 18 wt.%.

**Figure 2 nanomaterials-12-01522-f002:**
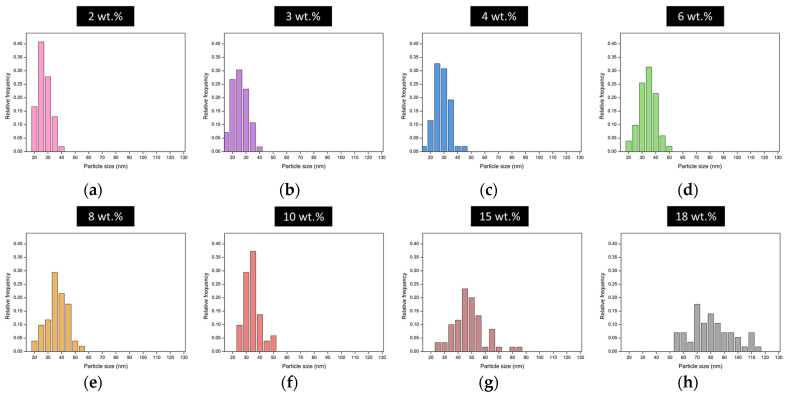
Particle size distribution for different aerogels produced using different catalyst contents, (**a**) 2wt.%, (**b**) 3 wt.%, (**c**) 4 wt.%, (**d**) 6 wt.%, (**e**) 8wt.%, (**f**) 10 wt.%, (**g**) 15 wt.% and (**h**) 18 wt.%.

**Figure 3 nanomaterials-12-01522-f003:**
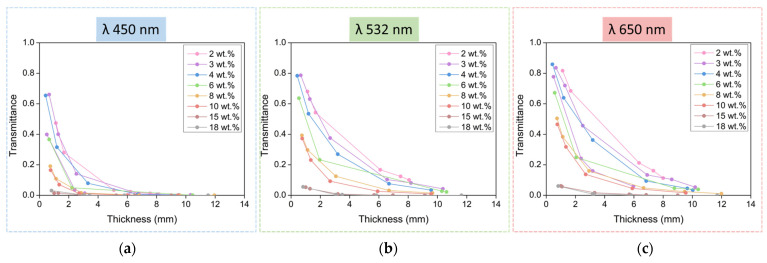
Transmittance dependence with sample thickness for materials produced with different catalyst contents at different wavelengths: 450 nm (**a**), 532 nm (**b**), and 650 nm (**c**).

**Figure 4 nanomaterials-12-01522-f004:**
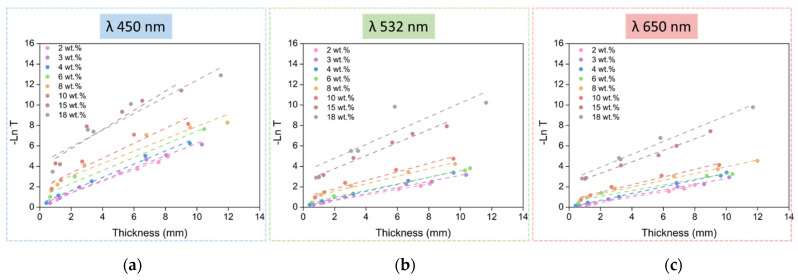
−ln(T) vs. thickness for all the aerogel samples understudy at three different laser wavelengths, (**a**) λ = 450 nm; (**b**) λ = 532 nm and (**c**) λ = 650 nm.

**Figure 5 nanomaterials-12-01522-f005:**
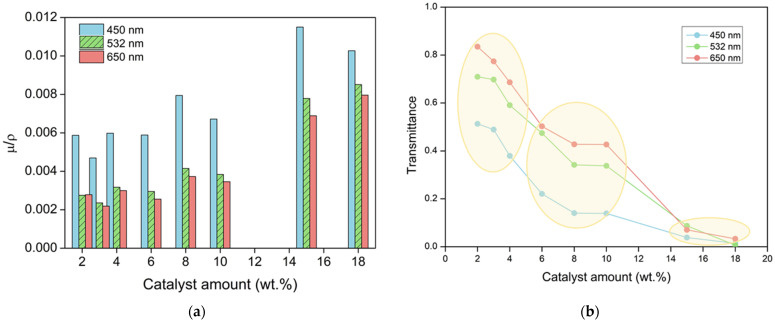
(**a**) Normalized attenuation coefficient for the aerogels containing each of the catalyst amounts and selected wavelengths. (**b**) Transmittance dependence with the catalyst amount for three different laser wavelengths for samples of 1 mm thickness. Numerical values can be found in Appendix A.

**Figure 6 nanomaterials-12-01522-f006:**
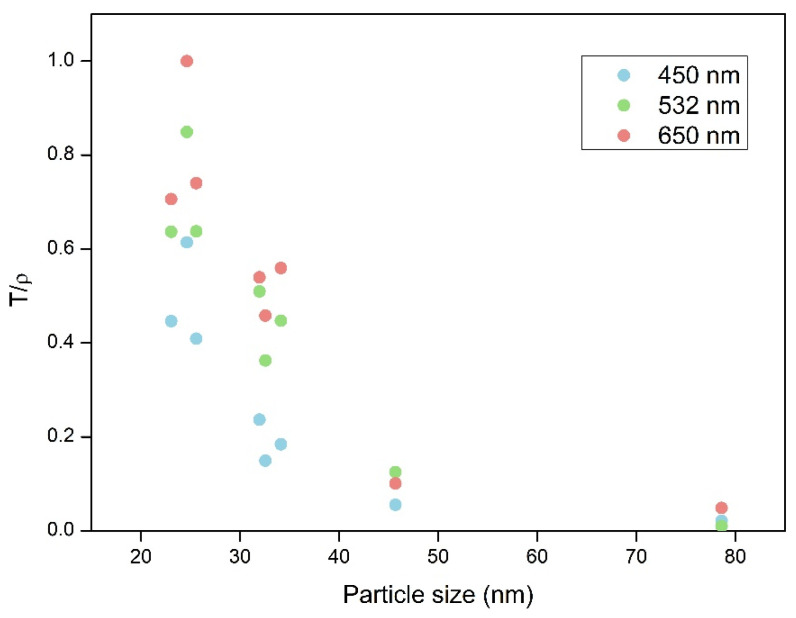
Dependence of normalized transmittance on the particle size of the final aerogels (1 mm thickness).

**Figure 7 nanomaterials-12-01522-f007:**
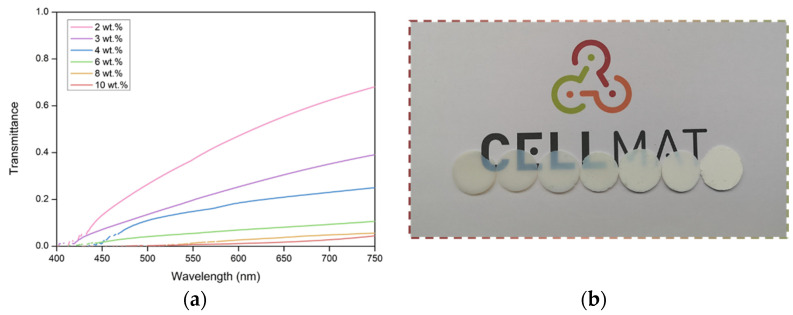
(**a**) UV–Vis transmittance measurements for all the samples under study. Measurement corrected for samples of 1 mm. (**b**) Optical micrographs of the samples from left to right; catalyst content and particle size increase in the following order: 2 wt.%, 3 wt.%, 4 wt.%, 6 wt.%, 8 wt.%, 10 wt.%, and 18 wt.%.

**Table 1 nanomaterials-12-01522-t001:** Structural parameters describing the aerogel structures.

Catalyst Amount (wt.%)	Density (g/cm^3^)	Porosity (%)	Particle Size (nm)	Average Pore Size (nm)
2	0.097 ± 0.001	91.68	24.6 ± 5.4	85.0
3	0.128 ± 0.023	89.08	23.0 ± 6.0	72.1
4	0.108 ± 0.015	90.77	25.6 ± 6.4	93.1
6	0.109 ± 0.011	90.73	32.0 ± 6.9	91.1
8	0.089 ± 0.001	92.38	34.1 ± 7.8	137.9
10	0.109 ± 0.001	90.72	32.6 ± 6.9	140.6
15	0.081 ± 0.001	93.03	45.7 ± 12.3	249.1
18	0.079 ± 0.003	93.23	78.6 ± 16.0	722.0

**Table 2 nanomaterials-12-01522-t002:** Average adjusted R^2^ values for the fitting of −Ln(T) as a function of L/λ^4^ or L/λ.

	L/λ^4^	L/λ
wt.%	R^2^	R^2^
2, 3, 4	0.994	0.962
6, 8	0.994	0.966
10	0.977	0.991

## Data Availability

The data presented in this study are available on request from the corresponding author.

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
