# Peer review of "Optical Properties of Polyisocyanurate–Polyurethane Aerogels: Study of the Scattering Mechanisms"

_nanomaterials, 2022, doi:10.3390/nano12091522_

Round 1
Reviewer 1 Report
In this article transparent PUR-PIR aerogels have been manufactured by using different catalyst amounts. The optical properties have been deeply analyzed. The dependence of the transmittance has been studied as a function of different parameters. Finally, the corresponding scattering mechanism was evaluated by analyzing the dependence of transmittance with a wide range of light wavelength (400 to 750 nm). The image is fuzzy, so the author is suggested to improve the image resolution. My detailed comments are as follows:
(1) Catalyst amount 6 and 10 wt.% have same density, porosity, and particle size, but their average pore sizes are very different. Similarly, their transmittance is relatively similar. With reference to other samples, no such difference has occurred. Can it be reasonably explained?
(2) Some grammar and spelling errors need to be checked carefully.
(3) In general, optical properties can be altered by controlling the morphology of aerogels. Is this property universal?
In conclusion, the manuscript is innovatory and interesting, it could be published with minor changes.
(4)Some writing problems are suggested to be modified by the author, such as subscript and superscript
(5)The image is fuzzy, so the author is suggested to improve the image resolution.
Reviewer 2 Report
Authors reported a systematic study of optical properties, particularly the transparency of PUR-PIR aerogels with respect to their particle size. The effect of particle size, wavelength, thickness have been discussed, and the manuscript is overly well-written. It might be published after addressing following questions.
- How to evaluate the transparency of carbon- and metal-based aerogels, e.g., DOI 10.1021/acsami.9b12918, DOI: 10.1126/sciadv.aaw4590?
- I would suspect the accuracy of acquiring the particle size by using the SEM images shown in Figure 1, where the resolution cannot meet the requirement.
- Figures 3-5 need to be re-formatted, since the figure legends are too small. The left plot of Figure 5 needs to re-draw. Additionally, please avoid to use a combination of red and green color.
- Why there is a sharp increase of normalized transmittance at the ~20-25 nm particle size?
Reviewer 3 Report
This study reflects on optical properties of polymeric PUR-PIR aerogels synthesized using different amounts of catalyst in the processing.
The general intention, "optical properties have been deeply studied" is laudable. A systematic approach has been attempted, and some aspects are well addressed. However, the actual implementation and performance – especially narrative and language – lack many standards.
#1. we found about 100 (one hundred) issues that highlighted in the manuscript, and we have attached the pdf with highlights to this report.
[despite being so many, we don't think this is exhaustive]
Most issues concern language, and include "deeply analyzed", improper choices like "a good transparency performance", a notorious "as a way", vague arguments using "may", misleading descriptions such as "slightly minimized", frequent "has been proved", as well as goodies such as "Measurements were made at ... (snip) ... at the University of Málaga (Spain) after being degassed under 132 high vacuum (24 hours at 50 °C).", where it remains a question whether the whole University was evacuated, or the measurements itself. Obviously, the samples/test tubes placed into the ASAP should have been mentioned.
That reading the manuscript for us is still possible – we have seen worse – is likely due to us originating in the same language family, which allows us to make some connections and jumps, although none are written down.
=> This must be improved significantly. The research, descriptions, argument, and conclusions must be comprehendible.
#2. standards describing experimental procedures are neglected.
Synthesis of aerogels is - at best – an abbreviated version of the earlier work. Even though, it should contain all steps necessary to reproduce (!) the synthesis. One example is the process of de-moulding and preparation for supercritical drying. How is this done? Any washing with liquid CO2 prior to drying? And while some parameters of the processing have been mentioned before (500 rpm ...) processing parameters of the drying are forgotten.
Control of temperature is important for the growth process of aerogels, and changes of 5C can easily hamper reproducibility of experiments. So, what are temperatures during synthesis?
The authors mention N2-sorption. Some arguments and explanations used explicitly the results of SSA and pore size distributions for the different samples. That implies that this data is important. However, no sorption data is shown to the reader, not even in the SI.
#3. the introduction is poor!
On one side, PUR-PIR aerogels are known for >30 years –– the authors note this in their original publication.
The sentence "Despite their huge interest, there are few examples of polymeric aerogels showing high light transmittance [19–22]" is just false. NASA holds patents on transparent polyimide aerogels –– non-silica. Multiple polymeric aerogels have been synthesized, some using cellulose and siloxane as cross linker like in doi.org/10.1016/j.nanoen.2018.03.029, others silicon-based polymeric precursors (example http://eprints-phd.biblio.unitn.it/1809/1/PhD_Thesis_E_Zera.pdf –– references therein).
Thus, there are already multiple (!) examples, non-silica and silicon (not silica!) based, for transparent aerogels. That needs to be improved to place the work into context.
Summary:
There was a previous original publication of the authors, from which methods and techniques are copied or "translated". It remains unclear as to whether samples characterized here are identical to those presented earlier, of if just the same approach has been used. The latter is, however, not comprehendible, since the descriptions lack details. Overall, without reading the original publication it is impossible to understand the actions in this research.
Some results [Section 3.1; Aerogel structure] appear to have been "borrowed" from the previous publication, or are a repetition of similar results –– hence, that doesn't appear to be new. It comes down to "Optical transmittance", which in this study is investigated using 3 different wave lengths – in contrast to just one wavelength of the previous publication – and significantly more details of the study are presented. That's of value, has merit, and should find its way to a journal. Interestingly, although transmittance of aerogel samples is the target, visual pictures of this phenomenon – like shown in the original publication – are not given. That's disturbing. Moreover, descriptions, reasonings, and conclusion that go along with it (the many "has been proved") are presented poorly.
Overall, the manuscript is not acceptable for publication. The content should be focused, the presentation overhauled, and the narration significantly improved.

Reviewer 4 Report
The manuscript entitled "Optical properties of polyisocyanurate-polyurethane aerogels: study of the scattering mechanisms" by Merrillas et al. submited to nanomaterials, presents the preparation of PUR-PIR aerogels and the study of their optical transmittance properties. These properties were correlated with the particle size and distribution. The overall subject is interesting, and the manuscript organization seems logical, with a rationale behind the preparation of the aerogels and their optical transmittance properties well supported by the results.
However, the authors should improve a few minor aspects.
1) In figure 1, the Information regarding to the meaning of the 2wt%, 3wt% and so on should be placed in the caption. It is much simpler for the reader to understand everything that is placed in the image just by looking at it.
2) The particle diameter was obtained by estimation from the SEM images. Is it possible to place SEM images with a higher magnification? This information should be placed in the Supporting Information.
3) In Figure 4, all the graphics are too small, and therefore, all the sample captions inside the graphics are difficult to read. Please increase the font.
Thank you
Round 2
Reviewer 1 Report
The author have revised the manuscript carefully according to the comments. Thus, I recommend this work can be published in Nanomaterials in present form.
Reviewer 3 Report
Some additions and clarifications have been provided. However, many “corrections” were done on the hoof, rushed, and without great thought.
There is still a lot (!) of effort necessary to improve the language. Tenses are often wrong and need to be corrected and adjusted [this becomes visible already in the abstract, where “is analyzed.” is followed by “It has been demonstrated”; later by “The results have showed ...”]. Please, this is found throughout the text including in the conclusions. It is really disturbing to read present, past, participle, past participle together in one section. Sometimes it is unclear, whether this indicates a sequence of actions – and is important – or just a mistake.
Moreover, a lot of unnecessary “filler” words and phrases are still used (the abstract starts with “in this work” – as if we would expect anything else; "It has been proved” –– since this is not a Math paper, the term "proved" defies the scientific method, and what understanding does this suggest of the authors?).
And, smile … “Measurements were made at – 196 °C in the range P/P0 = 0.05 – 0.30 after being degassed ..” is still in as well. That is just another indication that not much effort was made to make this a better paper.
A small program as simple as “Grammarly” would help the authors tremendously to alleviate most of the issues; even “Word” can help to correct the issues (albeit far less efficient).
